# Mind-Body Interventions as Alternative and Complementary Therapies for Psoriasis: A Systematic Review of the English Literature

**DOI:** 10.3390/medicina57050410

**Published:** 2021-04-23

**Authors:** Teodora Larisa Timis, Ioan Alexandru Florian, Daniela Rodica Mitrea, Remus Orasan

**Affiliations:** 1Department of Physiology, “Iuliu Hatieganu” University of Medicine and Pharmacy, 400012 Cluj-Napoca, Romania; doratimis@gmail.com (T.L.T.); rdmitrea@yahoo.co.uk (D.R.M.); rorasan@yahoo.com (R.O.); 2Department of Neurosciences, “Iuliu Hatieganu” University of Medicine and Pharmacy, 400012 Cluj-Napoca, Romania

**Keywords:** psoriasis, complementary and alternative medicine (CAM), acupuncture, balneotherapy, climatotherapy, cupping therapy, psychotherapy

## Abstract

Objective: Conventional therapeutic methods for psoriasis include topical and systemic drugs, phototherapy, and biologic agents. Despite the fact that these treatment methods, and especially biologic agents, are met with a considerable reduction in disease activity, they can sometimes be costly and are nonetheless accompanied by high risks of adverse events, ranging from mild to debilitating. Therefore, complementary and alternative medicine (CAM), especially mind-and-body interventions, such as acupuncture, psychotherapy, climatotherapy, and cupping may provide a cheaper and potentially beneficial outcome for these patients. Methods: We performed a systematic review of articles pertaining to acupuncture, cupping, psychotherapy and meditation, as well climatotherapy and balneotherapy in the management of psoriasis, by using the PubMED, Medline and Google Academic research databases and reference cross-checking. Results: 12 articles on acupuncture, 9 on dry or wet cupping, 27 concerning meditation, hypnosis or psychotherapy, and 34 regarding climate therapy or balneotherapy were found. Discussion and Conclusions: Currently, there is a lack of evidence in the English literature to support acupuncture as an effective alternative therapy for psoriasis, whereas cupping has been described in the majority of instances to result in Koebner phenomenon and clinical worsening. Stress management therapies such as psychotherapy, hypnosis, and meditation have shown promising results as complementary treatment methods. Climatotherapy and balneotherapy have already been proven as effective means of achieving clinical improvement in psoriasis. Further research is still needed to verify the usefulness of the lesser studied treatment methods.

## 1. Introduction

Psoriasis is a chronic inflammatory disorder that possesses decisive immunological and genetic elements, upon which several environmental factors may act, triggering the pathological cascade. Although it initially involves the skin, psoriasis leads to a substantial systemic impact, as well as a heavy psychological burden in many cases [1]. According to current data, it affects approximately 1–3% of the world population, presenting a first incidence peak between 15 and 20 years of age, followed by a secondary peak at 55 to 60 years [2]. In order to evaluate disease severity, various scores, such as the PASI (Psoriasis Area and Severity Index), or PGA × BSA (Physician Global Assessment and Body Surface Area) have been implemented, whereas the quality of life can be evaluated by the DLQI (Dermatology Life Quality Index) [3,4,5]. Presently, the PASI score is utilized for both the initial clinical evaluation of patients, and for their response to therapy. Although, until recently, PASI responses of 50 or 75 (denoting a 50% or 75% reduction in disease severity, respectively) were the primary aims for treatment, current methods strive for PASI 100, meaning a complete remission of cutaneous lesions [6,7,8]. These therapies include topically applied drugs such as dithranol, salicylic acid, corticosteroids, vitamin D analogues, or calcineurin inhibitors; phototherapy typically using either ultraviolet B (UVB), or a combination between psoralen and ultraviolet A (PUVA); systemic non-biologic therapies such as methotrexate (MTX), cyclosporin A, acitretin, apremilast, 6-thioguanine, or tofacitinib; and biologic agents that interfere with the pathological cascade of psoriasis near its outstart, these being TNF-α inhibitors (Adalimumab, Etanercept, Certolizumab, and Infliximab), the IL-12/23 inhibitor Ustekinumab, specific IL-23 inhibitors (Guselkumab, Risankizumab, and Tildrakizumab), and the IL-17 inhibitors (Brodalumab, Ixekizumab, and Secukinumab) [6,7,8,9,10]. The majority of modern therapies, whether focal, systemic or biologic, have a desired response of PASI 90 or 100, and are supported by a large volume of data as being regularly effective in ameliorating psoriasis. We, as practitioners, should always be informed and aware of recent developments, so that our selected treatment modality is optimal for each patient.

As conventional therapies are not always efficient in psoriasis, are high cost, and are sometimes associated with undesirable side effects, complementary and alternative medicine (CAM) may offer a safe and generally inexpensive substitute for some patients. Complementary medicine blends these approaches with conventional treatment, while alternative medicine is utilized instead of mainstream therapies. These are split into two groups: mind–body interventions (acupuncture, cupping, and meditation) and natural products, incorporating herbs, vitamins, and dietary supplements [11,12]. It is estimated that more than half of the patients with psoriasis seek CAM because of their dissatisfaction with the effects of conventional treatment methods [11,12,13], despite the fact that, until recently, CAM has been regarded as a non-evidence-based practice [13]. Hereafter, we present a systematic review of the studies on mind and body interventions performed for psoriasis that are available in the English literature.

## 2. Methods

We have performed a thorough search of the PubMed, Medline, and Google Academic databases for articles pertaining to mind-and-body interventions as CAM in psoriasis. No timeframe filters were utilized. For acupuncture and its related interventions, we used the keywords “acupuncture”, “moxibustion”, “needling”, “acupoint”, “puncture”, and “electrostimulation”. Regarding cupping therapy, the terms “cupping”, “moving cupping”, “dry cupping”, “wet cupping”, “Hijama”, or “Hijamah” were used. The keywords “psychotherapy”, “hypnosis”, “psychological support”, “support group”, and “meditation” were employed when searching for studies concerning stress management therapies. Lastly, the terms “balneotherapy”, “climatotherapy”, “climate therapy”, “Dead Sea”, or “Thalassotherapy” were employed for researching climate therapy articles. In conjunction with the aforementioned terms, “Psoriasis” was also used as a keyword for each individual search. Only articles with full text in English were considered. Furthermore, we only included studies performed on human subjects, regardless of prospective or retrospective nature, randomization, or number of participants (case reports were also encompassed). After full-text evaluation, additional references were verified and additional studies that did not appear at initial screening were included. Reviews and meta-analyses were excepted from the study itself, however they were employed for cross-reference checking.

## 3. Results

### 3.1. Data Acquisition

After the initial search and title screening, we obtained a total of 90 articles regarding acupuncture, 58 referring to cupping therapy, 146 pertaining to meditation and psychotherapy, and 298 related to climatotherapy. Subsequently, after eliminating duplicates, performing abstract screening and cross-reference checking, the total number of articles was reduced to 36, 15, 38, and 42 for each respective category. Full-text screening led to the removal of a further 24 acupuncture articles, which were animal studies, protocols for systematic reviews and prospective studies, systematic and non-systematic reviews, aside from articles written in Chinese or Korean that only had the abstract in English. Six cupping therapy articles which were written in Chinese and were reviews or opinions were also excluded. Review articles on psychotherapy, meditation and hypnosis, as well as studies on different dermatological diseases concerning these treatment methods or studies simply evaluating stress levels for psoriasis patients without any interventions were removed, these being 11 in number. Finally, eight articles on climate therapy, that addressed other dermatological ailments, were both systematic reviews or otherwise, or that simply evaluated epidemiological tendencies without outcomes, were discarded. The flowchart in Figure 1 summarizes the study inclusion/elimination process for each of the four treatment methods.

Regarding the articles on acupuncture and its analogous therapies, three were randomized controlled trials (RCTs), two were retrospective reports on non-randomized patients, one was a prospective observational report, and six were case reports of either one (four articles) or two (one article) patients. Excluding two articles, of which one was a prospective cohort of patients with various dermatoses, including psoriasis undergoing wet cupping [14], and the other a retrospective report of individuals who reported with secondary lesions after cupping [15], the remaining seven articles we obtained on cupping therapy were single-patient case reports. Fifteen articles on stress-reducing interventions were RCTs (one of which opted for a patient-preference randomization [16]); one was a non-randomized prospective trial, and one was a prospective cohort, whereas the remaining ten were case reports. As for climatotherapy and balneotherapy, we identified 6 prospective RCTs and 7 non-randomized trials, 12 single-arm prospective studies, and 7 retrospective studies (out of which 3 were comparative), one open, uncontrolled prospective trial [17], and one multicenter controlled cross-sectional study [18].

### 3.2. Patient Populations

In total, we identified 429 patients experiencing acupuncture for psoriasis; 254 males and 175 females. The average age was 42.43 years, the oldest patient recorded was 84 years, and the youngest was only 9 years [19,20]. The treatment periods lasted from 1 to 14 months, while the disease itself had a variable duration, from 4 to 20 years. A summary of patient characteristics can be viewed in Table 1.

Within the studies included, all 31 psoriasis patients treated via cupping were male. Ages ranged from 17 to 67 years, with an average of 36.14 years based on the ages provided. Duration of symptoms ranged from as little as 2 weeks to 10 years, while cupping treatment length varied from 5 days to up to 3 weeks. Table 2 presents a brief description of collected patients undergoing cupping therapy for psoriasis.

Adding together all the patients undergoing stress management therapies and psychological support, we obtained a total of 1085 individuals, of which 492 were female and 455 were male, based on the data provided. Patient age averaged at 43.78 years. Disease length at the time of treatment initiation had the largest range of all procedures analyzed, from the very day of symptom onset [21] to 64 years [16]. Duration of treatment also varied widely, from 6 weeks to as long as 4 years, depending on the exact procedure. More elaborate patient population details are presented in Table 3.

A total of 8498 patients suffering from psoriasis were enrolled in the studies of climate therapy and balneotherapy (9236 including control patients without psoriasis), and according to available and specified data, these numbers included 4702 males and 3675 females. The average age of patients undergoing these therapies across all studies was 48.02 years. Although no data was obtained regarding length of the disease, they varied from one to 60 years, the most common values being centered around 20–30 years. Treatment length was most commonly 3–4 weeks, although some balneotherapy interventions took 6 or 8 weeks [22,23,24], whereas some patients under climatotherapy followed this treatment for as little as 6 days [25]. It is worth mentioning that the study of Langeland and that by Wahl employed the same patient population based on their characteristics [26,27]. As such, this population was only considered once when performing calculations. Table 4 depicts the patient populations that have experienced balneotherapy and climate therapy.


medicina-57-00410-t001_Table 1Table 1Trials and case reports of patients undergoing acupuncture and similar therapies for psoriasis (arranged in order of references).Author, YearProcedureStudy DesignStudy/Treatment LengthPatients IncludedGenderMean Age (y)Average Duration of PsoriasisComparison TherapyDropout No. (%)Severity MeasurementOutcomeLiao and Liao, 1992 [19]AcupunctureRetrospective report9.1 sessions (average)61 Therapy25M/36F51.8416.26 yN/AN/AArbitrary, visualOn average, moderate improvement was achieved in psoriasis patients. However, no standard evaluation score was provided for accurate comparison.Ye et al., 2004 [20]Acupuncture plus Chinese herbsRetrospective report8 y study (30–50 days courses)80 Therapy44M/36F9–65NSN/AN/A
Therapy effectiveness of 91.3% (41 cured, 18 markedly improved, 14 improved).Jerner et al., 1997 [28]Electrostimulation via intramuscular acupuncture and ear acupunctureProspective RCT10 w54:35 Therapy19 Placebo31M/23F44 // 4820 y // 20 ysham (minimal) acupuncture2 (3.5%)PASIPASI lowered from 9.6 to 8.3 in the therapy group and from 9.2 to 6.9 in the placebo group, w/o statistically significant differences between the two.Pan et al., 2018 [29]Fire-needle therapy plus HXJDD and Vaseline creamProspective RCT4 w136:68 Therapy68 Control89M/47F44.92 // 45.1913.72 y // 13.41 yHXJDD and Vaseline cream alone15 (10%)PASI, DLQI, HAMA, CM syndrome scoreTherapy group did not attain significant improvement in PASI score compared to control group. However, significant differences were found between the two groups in alleviating CM syndrome and DLQI.Lu et al., 2012 [30]Auricular therapy with optimized Yinxieling FormulaProspective single-blind RCT8 w84:43 Therapy41 Control61M/23F38.98 // 38.58118.63 mo (9.9 y) // 136.79 mo (11.4 y)optimized Yinxieling Formula aloneN/APASI, DLQI, VAS, SDS, SASPASI lowered in both, yet significantly more in the therapy group. DLQI decreased in both groups, though not statistically significant and w/o statistical significance in difference between the two. SAS, SDS, VAS showed no statistical differenceJorge et al., 2016 [31]Ear acupunctureProspective observational report14 w or symptom resolution72M/5F3311.7 yN/A3 (30%)PASI5 of 7 patients presented complete disappearance of symptoms, two had significant recovery.Jeon YC, 2016 [32]Traditional Korean Medicine, especially Sa-Am acupuncture plus Chinese herbsCase report14 mo therapy1F253 yN/AN/ANSThe lesions on the patient’s back, abdomen right inner thigh, left foot, left side of the scalp had disappeared.Lee et al., 2019 [33]Traditional Korean medicine (acupuncture, herbal medicine, probiotics) and phototherapyCase report5 mo (case 1), 8 mo (case 2)21M/1F37/334 y // 7 yN/AN/APASI, VASPASI decreased from 7 to 1.2 (Case 1), and 23.2 to 2.2 (Case 2).Mahovic and Mrsic, 2016 [34]AcupunctureCase report1 mo1F4920 yN/AN/AArbitrary, visualImprovement of clinical aspect of psoriatic lesions.Zhu et al., 2011 [35]Needle acupunctureCase report1 mo1F3210 yN/AN/ANSAcupuncture induced Koebner phenomenon.Wu et al., 2013 [36]Acupuncture (for back pain)Case reportNot reported1M7320 yN/AN/ANSAcupuncture induced Koebner phenomenon.Zhu et al., 2017 [37]AcupunctureCase report5 mo1F4320 yN/AN/AArbitrary, visualImprovement of clinical aspect of psoriatic lesions.Abbreviations (in alphabetical order): CM, Chinese Medicine; DLQI, dermatology life quality index; F, female; GAD-7, 7-item Generalized Anxiety Scale; HAMA, Hamilton Anxiety Rating Scale; HXJDD, Huoxue Jiedu Decoction; M, male; mo, months; N/A, not applicable; NS, not specified; PASI, psoriasis area and severity index; RCT, randomized controlled trial; SAS, self-rating anxiety scale; SDS, self-rating depression scale; VAS, visual analogue scale; w, weeks; w/o, without y, years.
medicina-57-00410-t002_Table 2Table 2Clinical studies on patients undergoing dry and wet cupping therapy for psoriasis (arranged in order of references).AuthorYearProcedureStudy DesignStudy/Treatment LengthPatients IncludedGenderAge (y)Average Duration of DiseaseOutcomeEl-Domyati et al., 2013 [14]2013Wet cuppingProspective single-arm cohort4–6 sessions (2–4 w between sessions)50 (8 with psoriasis)MRange 17–67NSNo improvement was demonstrated in the psoriasis group and Koebner phenomenon appeared on 3 of the patients’ sites of cupping and incisions (upper and lower back).Sharquie et al., 2019 [15]2019Wet cuppingRetrospective report2 w24 (16 with psoriasis)MRange 25–40 (mean 32.5 for all cases)NSKoebner phenomenon appeared on the patients’ sites of cupping and incisions (back).Pavlov and Dimitrova, 2019 [38]2019Bloodletting and cuppingCase reportNS1M263 moKoebner phenomenon appeared on the upper and lower back, at the site of cupping (paravertebral).Malik et al., 2015 [39]2015Wet cuppingCase report3 w1M303 yPASI was initially 2, and 90% of lesions had disappeared after 3 sessions.Polat Ekinci and Pehlivan, 2000 [40]2020Wet cuppingCase report1 w1M398 moKoebner phenomenon appeared on the patient’s back at the site of the incisions and cupping (upper and lower back).Suh et al., 2016 [41]2016Dry cuppingCase reportNS1M383 yKoebner phenomenon appeared on the patients’ sites of cupping and incisions (back and buttocks) and aggravated psoriatic lesions on which cupping was performed.Tang L et al., 2021 [42]2021Dry cuppingCase report2 w1M352 wKoebner phenomenon appeared at the cupping area.Vender and Vender, 2015 [43]2015Dry cuppingCase report2 w1M454 moKoebner phenomenon appeared on the patient’s lower back at the sites of cupping.Yu et al, 2013 [44]2013Dry cuppingCase report5 d1M4010 yKoebner phenomenon appeared on the patient’s back, chest and abdomen, at the site of cupping.Abbreviations (in alphabetical order): d, days; M, male; mo, months; NS, not specified; w, weeks; y, years.
medicina-57-00410-t003_Table 3Table 3Trials and case reports of patients following psychotherapy, stress management, hypnosis, and meditation for psoriasis (arranged in order of references).AuthorProcedureStudy DesignStudy/Treatment LengthPatients IncludedGenderAge (y)Mean Duration of DiseaseComparison TherapySeverity MeasurementOutcomeFortune et al., 2004 [16]Psoriasis symptom management programPatient-preference randomized controlled trial6 w93:40 Therapy53 Control31M/62F25.2 // 20.218.5 y // 22.5 y (Range 3–64 y)Standard pharmacological care onlyPASISignificant difference on depression, anxiety, psoriasis life stress beliefs in severity of consequences of psoriasis, no/freq. of symptoms after therapy and follow-up.Abel et al., 1990 [21]Support group therapyCase reportseveral w/1 mo2F62/561 d/25 yN/AVisual aspect of lesionsImprovement in clinical condition for both patients.Price et al., 1990 [45]PsychotherapyRCT, pilot8 w23:11 Therapy12 Control12M/11F42.8/4617 y/25 yControlVAS, HADSVisible clinical improvement in within the active group. However, the VAS is a rather inaccurate measure of psoriasis, and this portion of the trial was handled in an open fashion.Schmid-Ott, 2000 [46]PsychotherapyCase report2 y1F4846 yN/ANSImprovement at the end of and 2.5 y after therapy.Shah and Bewly, 2014 [47]Psychological interventionCase report7 m1F4640 yN/APASIImprovement of psoriasis (baseline PASI of 24.8, then 0.6 after therapy). Shafii and Shafii, 1979 [48]Psychodynamic psychotherapyCase reportNS1MNSNSN/ANSImprovement of lesions and recovery of symptoms after psychotherapy.Tengattini et al., 2019 [49]Psychological interventionProspective, non-randomized trial6 mo86:33 Therapy53 ControlNS>18NSStandard carePASI, BSA, PSAB, GAD-7, DLQI, SF-12A statistically significant improvement in PASI and BSA was noticed in both groups at 6 months.Zachariae et al., 1996 [50]Psychotherapy (individual sessions)Prospective RCT12 w44:23 Therapy21 Control18M/26F38.7 // 39.5NSControl (no psychological or medical treatment)PASI, TSS, LDBFSlight but significant changes in TSS and LDBF in the therapy group, yet not in the control group. No differences between groups concerning PASI. The therapy group displayed significant reductions for all three psoriasis severity measures, w/o changes in the control group.Kabat-Zinn et al., 1998 [51]Mindfulness meditation-based stress reduction intervention plus UVB and PUVAProspective RCT13 w37:8 Therapy 110 Therapy 28 Control 111 Control 217M/20F4311.2 yUVB and PUVA alone (no audiotapes)Measure designed for this studyStatistically significant difference between tape and no-tape groups, attainment rate being approx. 3.8-times more likely in the former at halfway and clearing points.Frankel and Misch, 1973 [52]Hypnosis, psychotherapyCase report4 y1M3720 yN/ANSImprovement of psoriatic lesions.Kline et al, 1954 [53]HypnosisCase report11 w1F4520 yN/ASurface area and visual aspectImprovement of psoriatic lesions.Waxman, 1973 [54]HypnosisCase reportNS1F3820 yN/ANSImprovement of skin condition (score not mentioned).Tausk and Whitmore, 1999 [55]Hypnosis with active suggestions of improvementProspective RCT, single blind3 mo11:5 Therapy6 ControlNSNS>6 moNeutral hypnosis with no mention of their disease processPASI, VAS, Stanford Hypnotizability scaleIn the therapy group, the 3 highly hypnotizable patients had 81%, 43%, and 8% reductions in PASI at 3 mo. The 2 highly hypnotizable subjects in control had 31% and 18% amendment in PASI, whereas 2 of the moderately hypnotizable patients showed 13% and 22%, and 2 had a worsening of 36% and 18%, correspondingly.Gaston et al., 1991 [56]Meditation (T1)/meditation and imagery (T2)Prospective RCT12 w18:5 Therapy 14 Therapy 25 Control 14 Control 25M/13F34.313.7 ywaiting list (C1)/no treatment (C2)4-item scale by Lowe et alSignificant difference between therapy and control groups regarding mean psoriasis ratings after period, w/o any additional impact from imagery.Lazaroff and Shimshoni., 2000 [57]Medical Resonance Therapy Music plus normal therapyProspective RCT14 d30:20 Therapy10 Control13M/17 FRange 18–60NSNormal therapyStimulus to scratch and the degree of sicknessStimulus to scratch and degree of sickness were more reduced in therapy group (significance not specified).Paradisi et al., 2010 [58]Written emotional disclosure interventions and UVBProspective RCT4 mo40NS

UVB alonePASI, SAPASI, Skindex-29, DHQ-12Significant differences in Skindex-29 values between emotional writing group and others. Additionally, patients allocated to emotional writing had a longer period of remission after phototherapy.Vedhara et al., 2006 [59]Written emotional disclosure interventionProspective RCT12 w59:31 Therapy28 Control32M/27F5022 yControl writing interventionPASI, HADS, DLQI, POMSDisease severity and quality of life improved in both groups at the follow-up, w/o significant difference between therapy and control patients.Tabolli et al., 2012 [60]Pennebaker’s writing emotional disclosure plus educational interventionProspective RCT2 y202:97 Therapy102 Control124M/7847.94approx. 18 yEducational intervention alonePASI, SAPASI, Skindex-29, GHQ-12No significant differences in clinical or QoL between groups.Slight or no effect on the variables of interest by therapy.Fordham et al., 2014 [61]MBCT plus standard careProspective RCT8 w29:13 Therapy16 Control13M/16F41.1721.21 yStandard careSAPASI, DLQI, HADS, PSS-10Those in the MBCT group had a significant improvement concerning symptoms and QoL (DLQI) when compared to control cases.D’Alton et al., 2018 [62]MBCT (T1), MBSCT (T2), and MBSCT-SH (T3)Prospective RCT8 w94:25 Therapy 125 Therapy 222 Therapy 322 Control42M/52F49.8924.25 yStandard carePASI, DLQI, WHOQOL-BREFNo significant differences on psychological well-being, psoriasis symptom burden, or quality of life compared to TAU at post-treatment, 6- or 12-mo follow-up.van Beugen et al., 2016 [63]ICBT and standard careProspective RCT6 mo131:65 Therapy66 Control67M/64F52.69 // 53.4518.03 y // 15.16 yStandard carePASI, SAPASI, ISDL, RAND-36Larger improvements in ICBT compared to CAU regarding physical functioning and impact on daily activities, though not in psychological functioning, at 6-mo follow-up.Spillekom-Van Koulil, 2018 [64]ICBTCase report5 mo/6 mo21M/1F64/2628 y/1 yN/ANSImproved physical and psychological wellbeing, sustained at 6-mo follow-up.Bundy et al., 2013 [65]eTIPs—CBTProspective RCT6 w126:61 Therapy65 Control59M/67F45>16 yusual carePASI, HADS, DLQINo significant difference in PASI scores after intervention. Results may be limited by the large quantity of unavailable data.Seng and Nee, 1997 [66]Support group therapyProspective single-arm cohort7 w106M/4F37.515 yN/AKnowledge, acceptance and coping with the diseasePatients felt that the program enhanced their knowledge of psoriasis and increased their confidence in coping with the disease.Piaserico et al, 2016 [67]Biofeedback and CBT plus UVBProspective RCT8 w40:20 Therapy20 Control12M/28F49.717.7 yUVB alonePASI, GHQ-12, Skindex-29, STAIPatients undergoing therapy had a significant reduction in PASI score, from 9 at baseline to 3.8 and 2.5 at 4 and 8 w, correspondingly. Likewise, 65% of cases in the therapy group achieved PASI75 whereas only 15% of standard UVB patients did so at 8 w.Goodman, 1994 [68]Thermal biofeedbackCase report13 w1F5610 yN/ASkin temperatureAll 11 presenting psoriasis lesions vanished, and any new lesions appearing during therapy disappeared w/o visible scarring.Hughes et al., 1981 [69]Biothermal feedback and supportive psychotherapyCase report7 m1M312 yN/APsoriasis Rating Scale (developed by the authors)Marked dermatological improvement; inability to lower the temperature at the plaque site.Abbreviations (in alphabetical order): BSA, body surface area; C1, control 1; C2, control 2; CBT, cognitive behavioral therapy; DLQI, dermatology life quality index; eTIPs, electronic Targeted Intervention for Psoriasis; F, female; GAD-7, 7-item Generalized Anxiety Scale; GHQ-12, 12-item General Health Questionnaire; HADS, hospital anxiety depression scale; IBCT, internet-based cognitive behavioral therapy; ISDL, impact of skin disease on daily life; LDBF, laser Doppler blood flow; M, male; MBCT, mindfulness-based cognitive therapy; MBSCT, mindfulness-based self-compassion therapy; MBSCT-SH, self-help MBSCT; mo, months; N/A, not applicable; NS, not specified; PASI, psoriasis area and severity index; POMS, profile of mood states; PSAB, psoriasis skin appearance and bothersomeness; PSS-10; 10-items perceived stress scale; PUVA, photochemotherapy; QoL, quality of life; RAND-36, RAND-36 Health Status Inventory; RCT, randomized controlled trial; SAPASI, self-assessed PASI; SAS, self-rating anxiety scale; SDS, self-rating depression scale; SF-12, 12-item short form health survey; STAI, State-Trait Anxiety Inventory; T1, Therapy 1; T2, Therapy 2; T3, Therapy 3; TSS, total sign score; UVB, type B ultraviolet; VAS, visual analog scale; w, weeks; w/o, without; WHOQOL-BREF, The World Health Organization Quality of Life-BREF; y, years.
medicina-57-00410-t004_Table 4Table 4Trials and case reports of patients undergoing balneotherapy and climatotherapy for psoriasis (arranged in order of references).AuthorProcedureStudy DesignStudy/Treatment LengthPatients IncludedGenderMeanAge (y)Average Duration of PsoriasisComparison TherapySeverity MeasurementOutcomeBen-Amitai and David., 2009 [17]DSCOpen, uncontrolled prospective trial2 w178M/9FRange10–18NSN/APASI>75% improvement in six cases, and 50–75% amelioration (moderate) in five patients. At 6 mo. Follow-up, 12 patients were relapse-free, the other 5 presented mild relapse.David et al., 2005 [18]DSCMulticenter controlled cross-sectional study224 d (mean), 140 d (median)1198:460 Psoriasis738 Control261 M/199 F (P) 296 M/442 F (C)48 (P)47 (C)20 yBenign skin conditions including contact dermatitis and seborrhea (Control)NSElastosis, solar lentigines, poikiloderma, and facial wrinkles—significantly more common in psoriatic patients than controls, in relation with Dead Sea exposure time. No correlation with increased risk of malignant melanoma or NMSC in psoriatic patients.Eysteinsdóttir et al., 2014 [19]Balneotherapy with geothermal sea water and NB-UVBProspective RCT6 w68:22 GSW22 IT-GSW24 UVB39M/29F41/42.2/37.920 y/16.4 y/12.3 yBathing in GSW+NB-UVB vs. intensive treatment with GSW and NB-UVB vs. NB-UVB therapy alonePASI, DLQI, Lattice score, BMI% of patients attaining PASI 75 and 90 was significantly greater for both GSW and IT-GSW regimens than for NB-UVB monotherapy. Faster clinical and histological improvement, longer remission period and lower NB-UVB doses than for GSW-regimens than standard NB-UVB monotherapy.Baros et al., 2014 [23]BalneotherapyProspective RCT6 w60:19 T115 T226 C37M/27FNSNSStandard therapy vs. balneotherapy vs. standard therapy plus balneotherapyPASI, CRP, antistreptolysin O titer, iron, uric acidStatistically significant difference in remission length for patients treated with combination therapy and patients treated with standard therapeutic modalities, with best results for combination therapy.Gambichler et al., 2001 [24]Highly concentrated salt water balneotherapyProspective RCT, single-blind, left-right8 w104M/6F36 y2.5 yTap water balneotherapy on opposite elbowSeverity score pertaining to desquamation, erythema, and infiltration of the psoriatic plaques.Highly significant decrease in the baseline score, w/o significant difference between pre-treatment with salt water or tap water.Cohen et al., 2005 [25]DSCProspective single-arm cohort2 w (6–33 d)7040M/30FRange19–78 yRange 4–30 yN/APASI, BPSS (Beer Sheva Psoriasis Severity Score)75.9% reduction in PASI and 57.5% reduction in the mean BPSS.* Langeland et al., 2013 [26]Gran canaria climatotherapy and patient education programProspective single-arm cohort4 mo (3 w program)254152M/102F4724 yN/APASI, MHC-SFPositive mental health and health-related emotional distress recovered markedly after treatment. The longer the duration of psoriasis, and the presence of comorbidities, the greater the aforementioned improvement.* Wahl et al., 2015 [27]Gran Canaria climate therapyProspective single-arm cohort3 w254152M/102F4724 yN/APASI, SAPASI, Health Education Impact Questionnaire (heiQ)SAPASI score improved significantly, as well as self-management; at 3 mo. follow-up, only emotional distress and disease severity stayed significantly ameliorated.Hodak et al., 2003 [70]DSCProspective single-arm cohort4 w2718M/9FRange 24–73 y4–30 yN/APASI, quantitative histologic measuresAverage 81.5% decrease in PASI score, with complete therapeutic response in 13 subjects, marked in 5, moderate in 6, and slight in 2.Czarnowicki et al., 2011 [71]DSCRetrospective comparison4 w4025M/15FNSNSClimatotherapy alone vs. climatotherapy plus MTXPASI, BMI, BSADSC did not show better results in patients treated simultaneously with methotrexateEmmanuel et al., 2020 [72]DSCProspective single-arm cohort4 w1812M/6F52.234.2N/APASI, IGA, NAPSI, NAPPA, DLQI, EQ-5D-3L, BMIDSC led to PASI reduction of 88%, a mean decrease of 2.3 (76.7%) on the 5-IGA, and a QoL improvement as measured by DLQI and EuroQol 5D index.Emmanuel et al., 2019 [73]DSCRetrospectiveNS65M/1FNSNSN/APASI, BSA, histological specimens60.2% of new plaque areas reemerged within the site of former plaques, proved by histopathology.Even-Paz et al. [74]DSCProspective, non-randomized4 w4524M/21FNSNSSun exposure time of 3.0/4.5/6.0 h dailyPASI3 h of daily sun exposure at the Dead Sea, in two equal sessions from 09:00 and 14:00, respectively, were sufficient in treating psoriasis in July and August.Frentz et al., 1999 [75]DSCRetrospective nation-wide cohort6.1 y (0–22) treatment length NS1738872M/866F43NSN/APresence of NMSC or other cancers at follow-upOverall risk of cancer in patients undergoing DSC surpassed that expected in the general population, due to NMSC with an unusual distribution among body sites, age groups and sexes in these patients: young individuals and at multiple sites, numerous BCC being frequent in young women.Harari et al., 2011 [76]DSCRetrospective4 w treatment (study 2003–2007)605441M/164F48.0825.36 yDSC for <40 y age at onset of psoriasis vs. >40 y at onsetPASI, BSA74% of the patients <40 y at psoriasis onset had PASI 95, as opposed to 62% for >40 y. Therapeutic effect was inversely associated with the age of the patient at disease onset.Harari et al., 2007 [77]DSCProspective, single-arm cohort4 w6442M/22F41.416.8 yN/APASI, QoL VASAll subjects attained PASI 50, and 75.9% of cases reached PASI 75 after 1 mo of DSC, with a median remission length of 23.1 w and median therapeutic effect duration of 33.6 w. Younger patient age at therapy was correlated with a longer remission time.Harari et al., 2016 [78]DSCRetrospective report3–4 w719505M/214F51.2428.01 yN/APASI, BMI, body surface involvementPrevious DSC sessions were a positive predictor for an improved PASI, with a positive association between psoriasis duration (and younger age at onset) and PASI 90. More patients with photo skin type II (Fitzpatrick) achieved PASI 90 than other types.Harari and Shani., 1997 [79]DSCProspective, non-randomized trial4 w740428M/312FRange 10–72 yNSN/ABSA, rheumatologic indexPercentage of clearance was best (>72%) for patients staying in the sun at least 7 h daily; lack of psychologically support led to disease clearance in 68.9% of patients, compared to 75.8% of those supported. Previous DSC sessions, moderate to severe skin surface involvement, and coexistence of arthritis increased chances of psoriasis clearance.Kushelevsky et al., 1998 [80]DSCRetrospective comparison4 w80NSNSNSExposure to UVB exposure in other climate therapy regions (Sweden, Switzerland, Germany, Bulgaria, New Zealand)MEDMean UVB exposure dose in DSC stands among the lowest reported for psoriasis clearance. Despite comparable on a monthly basis, cumulative annual phototherapy exposure is noticeably higher than the UVB doses provided on a 4-w DSC session.Kushelevsky et al., 1996 [81]DSCProspective, non-randomized4 w688 (study 1)502 (study 2)1142 (study 3)320M/368F //238M/264F //583M/559F<10–>60 //>65NSN/ABSADuration of psoriasis of 40–49 y resulted in a higher clearance rate of 78.6%. The clearance rate in patients with disease onset at the ages of 10–19 y (77.8%) was superior to those in which the disease appeared 60 y (56.2%) // A mean decrease in systolic and diastolic blood pressures during DSC was noticed, regardless of sex and age, in all groups.Nissen et al., 1998 [82]DSC and bathingProspective, non-randomized4 w21:10 Psoriasis11 HealthyNSNSNSPsoriasis vs. healthy skin exposed to UVRIA for enkephalinTotal clearance of psoriasis at sample sites; average reduction of 21% in enk levels.Schewach-Millet et al., 1989 [83]DSCRetrospective study3 y (between 2 and 8 y)—3–4 w yearly19NSNSNSN/AHistological specimensIntensification of epidermal pigmentation when compared to pretreatment biopsy specimens in certain cases, w/o epidermal dystrophy or melanocytic atypia.Trøstrup et al., 2019 [84]DSCProspective single-arm cohort4 w4928M/21F51.86NSN/APASI, DLQI11/49 patients reached PASI; 10/49 presented increased PASI; age, sex, previous DSC sessions, and duration of observation period did not affect endpoints. DSC led to a significantly increased DLQI score in 60% of cases, even several mo follow-up, whereas 20% of patients presented a marked decrease in PASI.Bogdanov et al., 2012 [85]Climatotherapy and phototherapyProspective, non-randomized2 w93:45 psoriasis placate12 psoriasis palmoplantaris36 control55M/38F45NSNarrowband phototherapy aloneDLQICombined climatotherapy has a significantly better beneficial effect on the QoL of the patients with psoriasis placate than phototherapy alone.Golusin et al., 2015 [86]Rusanda Spa balneotherapy plus calcipotriolProspective RCT3 w60:30 Therapy30 Control28M/32F55.46 (T)41.73 (C)NSBalneotherapy plus calcipotriol vs. calcipotriol alonePASITherapy group showed a decrease in PASI score by 59.45%, whereas in the control group it was 39.34%. Topical calcipotriol associated with Rusanda Spa balneotherapy is more efficient than topical calcipotriol alone.Peroni et al., 2008 [87]Comano spa balneotherapyProspective, non-randomized1–2 w280:124 Therapy156 Control176M/104FRange18–85 yNSBPT (therapy) vs. balneotherapy alonePASI, SAPASI, Skindex-29, BSA1 w balneotherapy or BPT was enough to obtain statistically significant PASI score reduction. Both 2 w balneotherapy and BPT groups brought greater psoriasis amelioration with reduction in both PASI score and BSA.Péter et al., 2017 [88]BalneotherapyProspective single-arm cohort3 w8035M/45F63.7NSN/APASI, CRP levelAfter therapy, both PASI and CRP levels presented significant improvement.Tabolli et al., 2009 [89]Balneotherapy and BPTProspective, non-randomized2 w111:66 Therapy45 Control67M/24F>18 y25 < 10 y;81 > 10 yBPT vs. Comano balneotherapy aloneSAPASI, Skindex-29, GHQ-12, SF-36A decrease >50% after therapy in SAPASI 50 score was attained by 42% and 37% of patients in the BPT and BT groups, correspondingly. BPT group showed a statistically significant reduction in the number of GHQ-12 positive cases.Nilsen et al., 2009 [90]Gran Canaria climate therapyProspective single-arm cohort2 w2014M/6F47.2NSN/APASI, UV exposure, Spectral UVB (280–315 nm), UVA (315–400 nm) and CIE-weighted UVReduction in overall PASI score in all patients; no significant correlation between the reduction in psoriasis area severity index scores and UV doses;Martin et al., 2015 [91]Balneotherapy with selenium-rich spa water (La Roche-Posay)Prospective single-arm cohort3 w2932M/22F59NSComposition of skin microbial communities associated with unaffected and affected skin.PASI, PGA, Shannon Diversity IndexPASI scores decreased post-balneotherapy. Poor bacterial biodiversity was observed, with the bacterial communities being similar on both unaffected and affected adjacent skin. Family analysis identified, for the first time, Xanthomonadaceae belonging to Proteobacteria phylum and recognized as keratolytic, was linked with clinical improvement after 3 w balneotherapy.Pinton et al., 1995 [92]Balneotherapy with selenium-rich spa water (La Roche-Posay)Prospective single-arm cohort3 w9246M/46F47.3NSN/APASI, circulating soluble interleukin 2 receptor (sCD25) levelAverage PASI score decreased by 47 ± 4%. In total, 44 patients improved by >50%, 80 subjects responded with a mean decrease in the PASI score of 52-6 and 10 were stable with 10% improvement or less. Men responded markedly better than women. An association between PASI score decrease and the rise in the plasma selenium levels was noticed.Wang et al., 2020 [93]Balneotherapy plus Chinese herbal medicineProspective RCTNS190:97 Therapy93 Control103M/87FNSNSConsolidated CHM balneotherapy and NB-UVB (CTG) vs. unconsolidated (stopped after PASI drop to 1.8–2.0) (UTG)PASINo significant difference in PASI score between the two groups at the initiation and the termination of therapy. However, the mean remission length in CTG was 10.99 mo, significantly longer than UTG (7.94 mo).Tsoureli-Nikita et al., 2002 [94]Balneotherapy with Leopoldine spa waterProspective single-arm cohort4 w107M/3FRange 28–53 yNSLeopoldine water vs. double-distilled waterPASI, immunohistological studyAverage PASI improvement for Leopoldine spa water treated arms was 85.9%, whereas double-distilled water treated arms had 50.5% PASI improvement. Significant differences between cutaneous samples taken before and after 4 w of Leopoldine spa water therapy.Melandri et al., 2019 [95]Liman peloid baths and heliotherapy at Cervia spa, Emilia, ItalyProspective RCT2.5 w91:56 Therapy35 Control57M/24F52.3/57.9NSLiman peloid application followed by bath therapy vs. clay peloid mixed with tap waterPASICompared with the control group, there was a significant improvement in PASI score and fewer psoriasis recurrences in the therapy group, aside from a marked reduction in the topical use of cortisone and nonsteroid drugs.* Denotes the same patient population between the two studies, as indicated by the number of individuals recruited, gender distribution, mean age, the duration of the treatment program, and average length of disease [26,27]. Abbreviations (in alphabetical order): BCC, basal cell carcinoma; BMI, body-mass index; BPSS, Beer Shiva psoriasis severity score; BSA, body surface area; BPT, balneophototherapy; C, control; CRP, C reactive protein; CTG, consolidated therapy group; DLQI, dermatology life quality index; DSC, Dead Sea climatotherapy; enk, enkephalin; EQ-5D-3L, EuroQol−5 Dimensions−3 Levels (EQ-5D-3L); F, female; GHQ-12, 12-item General Health Questionnaire; GSW, geothermal spring water balneotherapy; H, healthy controls; 5-IGA, 5-point Investigator’s Global Assessment; IT-GSW, intensive geothermal spring water balneotherapy M, male; MED, minimal erythema dose; mo, months; MTX, methotrexate; N/A, not applicable; NAPPA, Nail Assessment in Psoriasis and Psoriatic Arthritis; NAPSI, Nail Psoriasis Severity Index; nm, nanometers; NMSC, non-melanoma skin cancer; NS, not specified; P, psoriasis; PASI, psoriasis area and severity index; PGA, physician global assessment; PUVA, photochemotherapy; QoL, quality of life; QoL VAS, quality of life visual analog scale; RCT, randomized controlled trial; RIA, radioimmunoassay; SAPASI, self-assessed PASI; SAS, self-rating anxiety scale; sCD25, circulating soluble interleukin 2 receptor; SDS, self-rating depression scale; SF-36, 36-item Short Form of the Medical Outcomes Study questionnaire; STAI, State-Trait Anxiety Inventory; T1, Therapy 1; T2, Therapy 2; UTG, unconsolidated therapy group; UV, ultraviolet; UVA, type A ultraviolet; UVB, type B ultraviolet; vs., versus w, weeks; w/o, without; y, years.


## 4. Discussion

### 4.1. Acupuncture

Acupuncture is a well-known Traditional Chinese Medicine (TCM) practice that has been successfully utilized for more than three millennia. It is generally considered a safe approach, with few side-effects, being accepted across the world for numerous ailments [96]. The exact origin of acupuncture is unknown, and a few styles have been defined, for example needling, moxibustion, cupping and acupressure. It is important to note that acupuncture is customarily used in combination with other therapeutic approaches of TCM, such as herbal remedies. In needling for psoriasis, there are several acupuncture points available in which, as the name implies, disposable needles are inserted into the skin to stimulate blood flow and reduce local inflammation though an as-of-yet imprecise mechanism [97]. A recent study performed on mice discovered that electroacupuncture, needling and fire needling was correlated with a lower local CD3+ T-cell population, as well as lower levels of substance P, neurokinin A, IL-17A, IL-1B, and IL-23p40 [96]. Acupoint stimulation should be implemented for a period of at least of six weeks in order to achieve therapeutic effect [98,99]. Some authors have reported a decreased recurrence rate of plaque psoriasis after acupuncture when compared to conventional medicine [96]. Still, not all authors agree on the efficacy of acupuncture, as data are scarce and, at least until recently, not always easily accessible to the researching community [98]. A recent meta-analysis on the use of acupuncture in psoriasis comprising 13 RCTs and a total of 1060 participants has shown that acupoint stimulation had a superior effect to the placebo (non-acupoint stimulation) [99]. However, the trials included in this meta-analysis clashed regarding the specific acupoints used, the exact number of stimulated points, as well as duration of acupuncture sessions. Furthermore, adequate blinding was realized in only two of the studies, thus making a proper comparison difficult. Nevertheless, according to the findings of these analyses, the acupuncture appears to provide benefits in the treatment of psoriasis irrespective of the stage of the disease, although it is challenging to ascertain the most advantageous technique [12,100].

Two RCTs did not show a significant improvement in PASI score when compared to a control procedure, either sham acupuncture [28], or oral Huoxue Jiedu Decoction and Vaseline cream alone [29], although the latter did demonstrate an amelioration in the quality of life. A third RCT revealed a significant improvement in PASI within both the treatment with auricular therapy plus optimized Yinxieling formula and control groups, yet more so in the former, whereas DLQI scores presented a non-significant decrease in these groups [30]. According to Jorge et al., ear acupuncture managed to result in the complete disappearance of psoriasis in five of their seven patients, while the remaining two presented marked recovery [31]. A retrospective study on 61 patients demonstrated moderate improvement in psoriasis patients, on average [19], while another such account of 80 patients observed a 91.3% treatment effectiveness, with 41 cured, 18 markedly improved, and 14 individuals improved [20]. However, no standard evaluation score was provided for accurate comparison. We identified four case reports totaling five patients that benefitted from clinical improvement after acupuncture [32,33,34,35], one even demonstrating the complete disappearance of lesions [32]. Two of these reports did not provide a severity score, and as such a more objective evaluation could not be performed [34,35]. There were also two case reports of patients presenting with Koebner phenomenon after acupuncture [36,37]. The Koebner phenomenon, as defined by the German dermatologist Heinrich Koebner (1838–1904), denotes the manifestation of isomorphic lesions at the sites of a cutaneous injury in an otherwise healthy skin [38,101]. It can occur in several dermatological afflictions, most commonly psoriasis, lichen planus, and vitiligo. Therefore, it is likely that interventions that involve damaging the dermis, such as acupuncture and cupping, may trigger this type of lesion in psoriatic patients.

Despite our rigorous search in this field of CAM, reference cross-check did not yield several of the cited articles. As such, we could not discuss some of the studies included in other reviews and meta-analyses. Regarding the use of acupuncture and its associated procedures, the majority of reports show a positive effect on the amelioration of psoriatic plaques. Several articles written in Chinese and not readily accessible to Western readers may provide further insight into the benefits of acupuncture in psoriasis. As of the writing of this review, a number of trials on the benefits of various acupuncture techniques in psoriasis are currently ongoing [97,102]. The reason for including acupuncture in our review of CAM methods for this disease is primarily because of its popular use in China and other Eastern countries.

### 4.2. Cupping Therapy

Cupping therapy is an ancient treatment method likened to acupuncture, also employed for various diseases. It has been described since antiquity, from the ancient Egyptians to the Chinese Han Dynasty, also being used in the times of Hippocrates and even to the early Islamic period [103,104,105]. Two types of cupping methods exist, namely dry and wet, also known as Hijama (or Hijamah) in Egypt and Arabic countries [39,40,41,42,43,44,106]. This treatment method creates a vacuum by placing glass suction cups directly on the skin of various body parts, mostly on the back, shoulders, buttocks or limbs [15,106]. The difference between dry and wet cupping stands in that the latter requires a skin incision either before or after performing the suctioning itself. Thus, it is believed that impurities in the blood and tissues can be drawn out, instead of simply transferred from one body site to another. The moving cupping method is a unique dry type of cupping that involves the application of lubricant (such as Vaseline) either to the treated body part or to the mouth of the glass cup and adsorbing the cup to the desired area. The physician then moves the glass cup manually across the skin in all directions while applying light force, thus producing flushing, heightened tissue blood flow, and in some cases even ecchymosis in the chosen treatment area [103]. This causes a local accumulation of antioxidant and anti-inflammatory products such as heme-oxigenase-1, carbon monoxide, biliverdin, and bilirubin, which also have antiproliferative and neruomodulatory effects [104]. Additionally, it was shown that cupping induces vascular endothelial growth factor (VEGF)-A expression in keratinocytes via the nitric oxide (NO)-mediated activation of hypoxia inducible factor (HIF)-1, thereby promoting angiogenesis [105]. It is thought that this method has the ability to increase skin tolerance and significantly improve its barrier function [104,105] and has already proven effective in the management of pain-related diseases, such as chronic low back pain or osteoarthritis [103]. A multicenter RCT trial is currently underway in China, aiming to determine the efficacy of moving cupping in the treatment of plaque psoriasis [103].

However, current evidence in the English literature is far from encouraging. The study of El-Domyati et al., which enrolled 50 patients with various dermatoses, including eight with psoriasis vulgaris, failed to show any improvement in psoriatic patients [14]. Moreover, three of these patients demonstrated Koebner phenomenon at the site of cupping, leading to the termination of therapy. Contrarily, all individuals with chronic idiopathic urticaria, 10 out of 11 of acne vulgaris patients (90.9%), and two out of nine with atopic dermatitis (22.2%), showed clinical improvement, whereas none of the patients with vitiligo presented any changes. In the report by Sharquie et al., 24 patients presented with on-site Koebner phenomenon after undergoing cupping, 16 (66.7%) of whom had been previously diagnosed with psoriasis [15]. Six other case reports demonstrated the Koebner phenomenon strictly on the regions receiving cupping therapy [40,41,42,43,44], and to the best of our knowledge, only the patient described by Malik et al. benefited from a reduction in disease severity after wet cupping [39]. Interestingly, only male patients were included in these reports, yet no explanation could be given for this reason. While we are aware that the Chinese literature holds several studies pertaining to the beneficial effects of cupping in psoriasis [107], these were either inaccessible to us or did not have an English full-text version. Thus, as it stands, there is little evidence in the Western literature to support cupping therapy as an effective or beneficial CAM management in psoriasis.

### 4.3. Psychotherapy, Stress Management and Meditation

Psoriasis is known to cause significant psychological distress, depression, feelings of stigmatization, and reduced health-related quality of life [108]. Moreover, stress has been recognized as a trigger factor in both the appearance and exacerbation of psoriasis, aggravating the cutaneous manifestations of the disease in more than half of the patients. Psychotherapy has been studied in several trials and individual case reports, with the results being promising [45,46,47], even as early as one millennium ago [48], but sometimes with little difference from the control groups receiving usual care or no treatment at all [16,49,50]. As of yet, the mechanism through which stress initiates or worsens this disease is unclear; however, some studies have shown marked improvement in the clinical state after psychotherapy. An RCT comparing phototherapy with and without listening to mindfulness-based stress reduction recordings during treatment sessions revealed that clinical improvement was achieved markedly faster in the meditation group [51]. The beneficial effects of hypnosis in psoriasis were evaluated in several case reports [52,53,54], as well as an RCT of 11 patients, which suggested that easily hypnotizable patients showed greater improvements in disease control [55]. Guided imagery, meditation, and cognitive-behavioral stress management was shown to offer moderate but statistically significant improvement when PASI, total sign score (TSS), and Doppler blood flow to psoriatic plaques were assessed [50]. Although the same trial did not yield a significant difference in PASI scores between the treatment group and the control group that did not receive any form of therapy. Meditation with or without imagery produced a marked clinical amelioration compared to no therapy at all, yet the mentioned study was limited by the small number of patients included [56]. The same effect can be observed when assessing Medical Resonance Therapy Music with standard care against standard care alone, although no statistical significance was specified [57].

In one prospective RCT of 40 patients, written emotional disclosure combined with UVB therapy led to a better clinical result and a longer period of time than the standard UVB treatment [58]. Nonetheless, two other RCTs on written emotional disclosure did show improvement in both treatment and control groups with control writing intervention (focusing on activities of the previous day) [59] and educational intervention, respectively [60], yet no significant difference between treatment and control groups was observed. Mindfulness-based cognitive therapy MBCT and its variants, mindfulness-based self-compassion therapy (MBSCT), and self-help MBSCT (MBSCT-SH), have yielded positive results, though mixed in comparison with treatment as usual [61,62]. Similarly, internet-based cognitive behavioral therapy (ICBT) managed to improve physical functioning and diminish the impact of psoriasis on everyday activities in patients presenting a psychological risk profile, and also enhanced and maintained psychological wellbeing [63,64]. Based on the RCT by Bundy et al., a web-based online electronic Targeted Intervention for Psoriasis (eTIPs), also a form of cognitive-behavioral therapy, did not achieve a significantly different result when compared to standard care, yet, as the authors mentioned, the results were constrained by a large quantum of missing data [65]. Support group therapy may have the benefit of both clinical improvement and enhancing the patients’ knowledge and ability to cope with the disease [21,66]. As stated by Piaserico et al., biofeedback and cognitive-behavioral therapy and UVB therapy resulted in a significant reduction in psoriasis severity, as quantified by PASI, in addition to a higher percentage of patients achieving PASI 75 response at 8 weeks in comparison to patients receiving only UVB therapy [67]. Thermal biofeedback led to the complete disappearance of all previously existing psoriasis lesions, as well as the disappearance without scarring of any new ones occurring during treatment in one reported case [68]. In another similar description, thermal biofeedback in conjunction with supportive psychotherapy managed to markedly ameliorate dermatological signs [69]. Although promising, these studies are few in number, are small in size, and some of them are steadily becoming outdated. Furthermore, despite no obvious risk tied to meditation or hypnotherapy, some argue that there is little evidence to support them as financially justifiable treatment methods [109]. Nevertheless, this may prove an advantageous complementary treatment method of psoriasis in the future.

Considering that psoriasis is stress-mediated, it stands to reason that psychotherapy and interventions that focus on stress reduction might be beneficial for these patients. The results of the majority of studies are promising; however, not conclusive. The remarkable variation in therapy length may be due to the heterogeneity of therapies included in this group, as well as individual characteristics and requirements of each patient. It is important to notice that these therapies are mostly used in conjunction with treatment as usual and should not be viewed as a replacement to standard care.

### 4.4. Balneotherapy and Climatotherapy

Balneotherapy and Climatotherapy denote already established CAM treatment methods in moderate-to-severe psoriasis, having been proven effective in the short-term clearing and remission induction across several studies [18,70,110,111]. The most common destination for climate therapy is the Dead Sea, with 18 out of the total of 34 studies included having been conducted there [17,18,25,70,71,72,73,74,75,76,77,78,79,80,81,82,83,84]. This treatment method implies spending several weeks at the Dead Sea, bathing in its waters and lying in the sun. The Dead Sea is located on the lowest point on the landmasses of Earth, at approximately 400 m below sea level, possessing the highest concentration of salt of any natural body of water. It boasts exceptional climatic properties, which are beneficial for a wide variety dermatological conditions, specifically for psoriasis. The efficacy of Dead Sea climatotherapy is probably the result of a mixture between the anti-inflammatory effects of stress reduction, the anti-proliferative and keratolytic effects of local minerals, and the particular UV characteristics at that latitude [18,70,110,111]. More precisely, UVA and longer wavelength beneficial UVB rays are found at the site of the Dead Sea, whereas shorter erythrogenic UVB rays are generally filtered [11]. Severe adverse events following this type of therapy are rare. According to David et al., climatotherapy at the Dead Sea for psoriasis patients was more frequently associated with elastosis, solar lentigines, poikiloderma, and facial wrinkles than in control patients, also displaying an exposure-dependent response [18]. Additionally, the same study concluded that Dead Sea climate therapy was not correlated with a heightened risk of developing melanoma or nonmelanoma skin cancer in these patients. Another retrospective study concluded that some of these patients present an increase in epidermal pigmentation when compared to pretreatment biopsy specimens, although there were no epidermal dystrophies or melanocytic atypia reported [83]. Contrariwise, consistent with the findings by Frentz et al., the overall risk of skin malignancies (especially non-melanoma skin cancer) in patients undergoing this therapy was higher than estimated for the general population [75]. The body surface distribution of cutaneous cancers favored multiple sites, and typically affected younger individuals, especially women. Reoccurrence of psoriatic lesions at previous sites can occur after a given period of time following Dead Sea climate therapy [73].

Several prospective cohorts demonstrated a significant decrease in PASI scores in patients following Dead Sea climatotherapy, the majority of individuals achieving a PASI 75 response or more [25,70,72,74,77,79,84]. Furthermore, some of these studies also showed an improved quality of life, as measured by DLQI [72,84]. Both Kushelevski and Harari reported a higher clearance rate of lesions in patients with early-onset psoriasis and those with a longer duration of the disease [76,77,78,81]. Currently, the influence of the number of previous climate therapy stays on clinical amelioration has not been definitively established [78,84]. Another ambiguity is the daily exposure to sun needed to effectively treat psoriasis, on one hand Even-Paz et al. stating that 3 h divided in two equal sessions from 9 AM and 2 PM were sufficient when compared to 4.5 and 6 h per day [74], and on the other, Harari and Shani reporting that the best results were obtained in patients staying in the sun at least 7 h daily [74]. Patients additionally receiving systemic therapies such as methotrexate might not demonstrate better results than those undergoing climate therapy alone [71].

Other locations that have climatotherapeutic or balneotherapeutic (bathing in hot springs) effects are in the Black Sea, Nord Sea, Baltic Sea, Canary Islands, Kangal Hot Springs in Turkey, or the Blue Lagoon in Iceland, but the evidence regarding each of these sites is lacking when compared to the Dead Sea [11,18,70,110,111]. Balneotherapy alone or in combination with standard treatment or phototherapy has repeatedly proven to be more effective than standard treatment alone [23,85,86,87,88,89]. Geothermal sea water balneotherapy and narrowband UVB (NB-UVB) light therapy is apparently more effective in attaining clinical and histological amelioration, results in longer remission time and allows for lower UV doses than NB-UVB therapy alone [21]. Gran Canaria climate therapy has also shown potential in decreasing psoriasis severity, as well as promoting mental health and improving health-related emotional distress [25,26,90]. Balneotherapy in the selenium-rich waters of La Roche-Posay also has the potential of reducing severity [91,92]. Moreover, as concluded in the study of skin microbiome composition in these patients, the Xanthomonadaceae family associated with Proteobacteria phylum, and recognized as keratolytic, was linked to clinical amelioration after a 3-week balneotherapy treatment [91].

Consolidated balneotherapy supplemented with Chinese herbal medicine led to a notably longer remission time than the unconsolidated form, which was stopped after PASI dropped to 1.8–2.0 [93]. Leopoldine spa water balneotherapy showed a marked and statistically significant improvement when compared to double-distilled tap water treatment on the opposite arms of the same patients [94]. However, in another similar study, no significant difference was noticed between highly concentrated salt water and simple tap water balneotherapy [24]. Liman peloid application and bath therapy was also associated with benefits to severity score, time to recurrence, and a reduction in topical drug use than clay peloid and tap water [95].

This treatment form presented the most homogeneous treatment length, usually at around 3–4 weeks, although extremes of 6 days and 8 months were also noticed. Additionally, the average age of patients following these therapies was higher than for the other treatments, though this could be a simple incidental observation. The evidence on hand supports balneotherapy and climate therapy as complementary therapies for psoriasis, capable of even inducing remission. A special recommendation should be given to patients with early-onset psoriasis and those with a longer disease duration, while caution must be taken for immunocompromised individuals or those with a history of skin malignancies.

## 5. Limitations

The major limitation of this review is the exclusion of numerous articles written in Chinese or Korean that were unfortunately inaccessible and could have provided valuable insight into the effects of CAM methods in psoriasis. It is possible that the conclusions drawn would have differed considerably, had these studies also been incorporated. We are aware that the English literature is lacking in fields such as acupuncture and cupping therapy, and so practitioners are specialized in these interventions in Western countries, as opposed to those in Asia. Furthermore, several articles that we tried to verify through reference cross checking could not be found on any of the search engines we utilized. Another limitation is that some of the conclusions, especially concerning acupuncture and cupping, are drawn from single-patient case reports, which do not hold the same statistical significance as randomized controlled trials.

## 6. Conclusions

CAM, with reference to stress management interventions, balneo-, and climatotherapy, in conjunction with usual care, has the potential to help psoriasis patients reach remission faster and for longer periods of time compared to standard therapy alone. Concerning acupuncture, despite being popular in China even for the management of psoriasis, the English literature is as of yet inconclusive, whereas no evidence supports cupping therapy as being safe or effective, but potentially harmful due to the extremely high incidence of subsequent Koebner phenomenon. Therefore, these latter two options are not viable alternative treatment methods based on current available evidence. We emphasize that standard therapies, such as topical, systemic, and biologic agents, should always be considered before attempting CAM methods, since there is a larger body of evidence to support the effectiveness of these treatments. Furthermore, additional research in the form of prospective trials should be performed before establishing acupuncture and cupping as beneficial in psoriasis management.

## Figures and Tables

**Figure 1 medicina-57-00410-f001:**
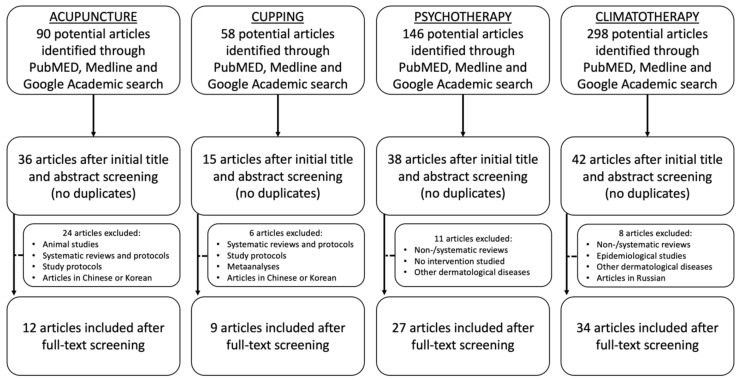
Analytical flow-chart of the employed article selection and elimination processes.

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
