# Peer review of "Mind-Body Interventions as Alternative and Complementary Therapies for Psoriasis: A Systematic Review of the English Literature"

_medicina, 2021, doi:10.3390/medicina57050410_

Round 1

Reviewer 1 Report

An unusual review about the use of "complementary" therapies in psoriasis. I suggest the author have the text checked by a native English language speaker, as some expressions do not fit well. The paper is however very complete, given the high number of studies included. Only minor queries:

"Presently, the PASI score is utilized for both initial clinical evaluation of patients, and for their response to therapy. Although up until recently a PASI response of 50 or 75 (denoting a 50% or 75% reduction of disease severity, respectively) were the primary aim for treatment, current methods strive for PASI 100, meaning a complete remission of cutaneous lesions. The majority of modern therapies, whether focal, systemic or biologic, have a desired response of PASI 90 or 100, and we as practitioners should always be informed and aware of recent developments, so that our selected treatment modality is optimal for each patient." these long paragraph needs some referrals, andalso probably needs a specification about topical drugs(such as corticosteroids and vitamin D derivatives) and systemic treatments (biologics anti TNF alpha, IL17 and IL23 )  here some articles you could consider:doi: 10.1371/journal.pone.0241575. doi: 10.1111/dth.14504.   doi: 10.1111/dth.13185.

Thank You

Author Response

Cover letter for: medicina-1176657

Mind-Body Interventions as Alternative and Complementary Therapies for Psoriasis: A Systematic Review of the English Literature

Esteemed editor and reviewers,

We, the authors, would like to express our gratitude for your kind remarks and suggestions for our manuscript. We hope that the changes made are to your expectations. If, however, you would like further changes, we are more than happy to comply. Below, you will find a point-by-point response to each suggestion, according to the reviewer.

Reviewer 1

An unusual review about the use of "complementary" therapies in psoriasis. I suggest the author have the text checked by a native English language speaker, as some expressions do not fit well. The paper is however very complete, given the high number of studies included.

Response: We are grateful for your remarks and suggestions. We have considered this recommendation and have decided on submitting our manuscript for the English Editing Service provided by MDPI.

Only minor queries:

"Presently, the PASI score is utilized for both initial clinical evaluation of patients, and for their response to therapy. Although up until recently a PASI response of 50 or 75 (denoting a 50% or 75% reduction of disease severity, respectively) were the primary aim for treatment, current methods strive for PASI 100, meaning a complete remission of cutaneous lesions. The majority of modern therapies, whether focal, systemic or biologic, have a desired response of PASI 90 or 100, and we as practitioners should always be informed and aware of recent developments, so that our selected treatment modality is optimal for each patient." these long paragraph needs some referrals, and also probably needs a specification about topical drugs(such as corticosteroids and vitamin D derivatives) and systemic treatments (biologics anti TNF alpha, IL17 and IL23 )  here some articles you could consider: doi: 10.1371/journal.pone.0241575. doi: 10.1111/dth.14504.   doi: 10.1111/dth.13185.

Thank You

Response: We appreciate for pointing this out, we have included your three recommended articles and two additional others on standard therapies into the references. We have also mentioned some of the representatives of these treatments (topical, phototherapy, systemic and biologic) and the fact that their efficiency is supported by a large volume of data (lines 48-59).

Reviewer 2 Report

This review provides an interesting and thorough summary of CAM options for psoriasis, with some of the options being more evidence-based than the others. The only critique I would have is perhaps placing more of an emphasis on the fact that while these are potential alternate therapies, there are many, many evidence-based, effective therapies -- topical, oral, and biologic -- for psoriasis that can and should be considered prior to undertaking CAM options. I also question the logic of including acupuncture and cupping in the review if they are not considered viable CAM options at this time due to lack of evidence and indeed evidence of harm through increaesd risk of Koebnerization; I would further emphasize that these are not viable treatment options at this time. 

Author Response

Cover letter for: medicina- 1176657

Mind-Body Interventions as Alternative and Complementary Therapies for Psoriasis: A Systematic Review of the English Literature

Esteemed editor and reviewers,

We, the authors, would like to express our gratitude for your kind remarks and suggestions for our manuscript. We hope that the changes made are to your expectations. If, however, you would like further changes, we are more than happy to comply. Below, you will find a point-by-point response to each suggestion (in italic), according to the reviewer.

Reviewer 2:

This review provides an interesting and thorough summary of CAM options for psoriasis, with some of the options being more evidence-based than the others. The only critique I would have is perhaps placing more of an emphasis on the fact that while these are potential alternate therapies, there are many, many evidence-based, effective therapies -- topical, oral, and biologic -- for psoriasis that can and should be considered prior to undertaking CAM options. I also question the logic of including acupuncture and cupping in the review if they are not considered viable CAM options at this time due to lack of evidence and indeed evidence of harm through increaesd risk of Koebnerization; I would further emphasize that these are not viable treatment options at this time..

Response: We thank you for your kind remarks and recommendations. We have added a few sentences mentioning the standard treatment methods in the Introduction, as suggested by Reviewer 1, also adding that they are supported by a large volume of data (lines 48-59). Moreover, we have emphasized in the conclusion the fact that the current evidence does not yet support acupuncture and cupping as complementary and alternative treatment methods for psoriasis, and that standard therapies should be always considered before CAM (lines 852-859). We have also added our reasoning for including acupuncture in this review, that being that it is a popular treatment method even for psoriasis in China (lines 286-288).

Reviewer 3 Report

Interesting and valuabe review concerning an undervalued issues as  complementary and alternative medicine (CAM) which in conjunction with usual care has the potential to help psoriasis patients reach remission faster  than compared to standard therapy alone. The paper is thoroughly prepared, well written. The authors rightly underlined the limitations of different CAM methods. They also stressed the need for further research in the form of prospective trials that should be performed before establishing acupuncture and cupping as beneficial in psoriasis management.

Author Response

Cover letter for: medicina- 1176657

Mind-Body Interventions as Alternative and Complementary Therapies for Psoriasis: A Systematic Review of the English Literature

Esteemed editor and reviewers,

We, the authors, would like to express our gratitude for your kind remarks and suggestions for our manuscript. We hope that the changes made are to your expectations. If, however, you would like further changes, we are more than happy to comply. Below, you will find a point-by-point response to each suggestion (in italic), according to the reviewer.

Reviewer 3:

Interesting and valuable review concerning an undervalued issues as complementary and alternative medicine (CAM) which in conjunction with usual care has the potential to help psoriasis patients reach remission faster  than compared to standard therapy alone. The paper is thoroughly prepared, well written. The authors rightly underlined the limitations of different CAM methods. They also stressed the need for further research in the form of prospective trials that should be performed before establishing acupuncture and cupping as beneficial in psoriasis management.

Response: We sincerely thank you for your kind remarks. We hope that you will agree with the modifications made for the revision of our manuscript.